# Enhanced Antioxidant Activity and Reduced Cytotoxicity of Silver Nanoparticles Stabilized by Different Humic Materials

**DOI:** 10.3390/polym15163386

**Published:** 2023-08-12

**Authors:** Maria V. Zykova, Alexander B. Volikov, Evgeny E. Buyko, Kristina A. Bratishko, Vladimir V. Ivanov, Andrey I. Konstantinov, Lyudmila A. Logvinova, Dmitrii A. Mihalyov, Nikita A. Sobolev, Anastasia M. Zhirkova, Sergey V. Maksimov, Irina V. Perminova, Mikhail V. Belousov

**Affiliations:** 1Pharmaceutical Faculty, Siberian State Medical University, 634050 Tomsk, Russia; buykoevgen@yandex.ru (E.E.B.); kr-1295@mail.ru (K.A.B.); ivanovvv1953@gmail.com (V.V.I.); ludmila_logvinova@mail.ru (L.A.L.); diman021999@gmail.com (D.A.M.); mvb63@mail.ru (M.V.B.); 2Department of Chemistry, Lomonosov Moscow State University, Leninskiye Gory 1-3, 119991 Moscow, Russia; ab.volikov@gmail.com (A.B.V.); konstant@med.chem.msu.ru (A.I.K.); n.a.sobolev@outlook.com (N.A.S.); zhirkova_am@mail.ru (A.M.Z.); irber@yandex.ru (S.V.M.); iperm@med.chem.msu.ru (I.V.P.)

**Keywords:** humic substances, silver nanoparticles, antioxidant activity, cytotoxicity, biomedical applications

## Abstract

The current article describes the biological activity of new biomaterials combining the “green” properties of humic substances (HSs) and silver nanoparticles. The aim is to investigate the antioxidant activity (AOA) of HS matrices (macroligands) and AgNPs stabilized with humic macroligands (HS-AgNPs). The unique chemical feature of HSs makes them very promising ligands (matrices) for AgNP stabilization. HSs have previously been shown to exert many pharmacological effects mediated by their AOA. AgNPs stabilized with HS showed a pronounced ability to bind to reactive oxygen species (ROS) in the test with ABTS. Also, higher AOA was observed for HS-AgNPs as compared to the HS matrices. In vitro cytotoxicity studies have shown that the stabilization of AgNPs with the HS matrices reduces the cytotoxicity of AgNPs. As a result of in vitro experiments with the use of 2,7-dichlorodihydrofluorescein diacetate (DCFDA), it was found that all HS materials tested and the HS-AgNPs did not exhibit prooxidant effects. Moreover, more pronounced AOA was shown for HS-AgNP samples as compared to the original HS matrices. Two putative mechanisms of the pronounced AOA of the tested compositions are proposed: firstly, the pronounced ability of HSs to inactivate ROS and, secondly, the large surface area and surface-to-volume ratio of HS-AgNPs, which facilitate electron transfer and mitigate kinetic barriers to the reduction reaction. As a result, the antioxidant properties of the tested HS-AgNPs might be of particular interest for biomedical applications aimed at inhibiting the growth of bacteria and viruses and the healing of purulent wounds.

## 1. Introduction

Research on metal nanoparticles has been emphasized over the past decade due to the controllable size and shape, ease of synthesis, and optical properties of these compounds [1]. Today, metal nanoparticles are used in all fields of science, including physics, chemistry, computer science, and biomedicine [1,2,3].

Currently, there is an acute need for new forms of pharmaceutical objects. Nanomaterials exhibit very high surface-area-to-volume ratios and surface energies. Attempts have been made to combine nanomaterials and modern antimicrobial drugs which exhibit synergism and improve the effect several times over [4]. In addition, new therapies based on targeted drug delivery [5] and nanoparticle-activated photothermal therapy are being actively investigated [6], as well as their potential for use in tissue and tumor imaging applications [7].

The most studied and promising candidates for unconventional and effective applications in pharmaceutical sciences, cosmetic products, antimicrobial coatings, and food packaging are silver nanoparticles (AgNPs). The particular interest in AgNPs for use in biomedical applications is based on their extensive antibacterial, antimycotic, and antiviral properties, biocompatibility, and efficacy against multidrug-resistant microorganisms [8,9].

Processes for the synthesis of AgNPs using chemical or physical methods (by microwave or γ-radiation, thermal decomposition, and chemical reduction) often require the use of toxic reducing and stabilizing agents (hydrazine hydrate, sodium borohydride, ethylene glycol) [10] and involve other environmental and biological risks [11]. Therefore, “green synthesis” or “biogenic synthesis” is recognized by experts as an eco-adaptive approach to create a variety of nanomaterials and is now being used to produce nanomaterials for biomedical applications [11,12,13,14]. The biosynthesis of nanoparticles can be achieved by the reduction of metal cations using extracts of plants, yeast, algae, lichens, fungi, and bacteria as starting matrices.

One of the most convenient and environmentally friendly ways to synthesize metal nanoparticles is the reduction of the metal-containing precursor in aqueous medium in the presence of a stabilizer for the resulting nanoparticles [15,16] (polysaccharides [17], polyphenols [18,19], etc.).

In addition to the antibacterial activity of AgNPs, other types of activity are possible, depending on the nature of the stabilization matrix and its properties. For example, there is evidence for the antioxidant activity (AOA) of silver nanoparticles stabilized with Achyranthes roughii leaf extract [20].

Humic substances (HSs) are supramolecular ensembles of oxidative degradation products of biomacromolecules and are very promising ligands (matrices) for AgNP stabilization [21]. Engagement in this process involves significant constituents such as plant lignin and its derivatives, polysaccharides, proteins, lipids, nucleic acids, and other natural biomacromolecules [22]. This intricate composition imparts HSs with a combination of both aromatic scaffolding and aliphatic segments within their molecular frameworks. Humic substances can be divided based on their solubility into humic acids (HAs), which are not soluble at pH values below 2, and fulvic acids (FAs), which are soluble across the entire pH range [23]. The unique chemical feature of HS is their extreme structural heterogeneity, which contributes to their resistance to biodegradation [21,24]. During the investigation of the HS molecular structure using high-resolution Fourier transform ion cyclotron resonance (FTICR) mass spectrometry, several tens of thousands of molecular formulae in the HS structure were identified [25]. The polyfunctionality of HSs acts as a fundamental property. The structural composition of HSs encompasses a diverse array of functional groups including, but not limited to, phenolic−OH, quinones, hydroxyls, methoxyls, and carboxyls. It provides their ability to participate in ligand exchange and heterogeneous processes, forming a variety of intra- and intermolecular bonds. These properties can be taken together to determine the redox, chelating, and protolytic potency of HSs. HSs are characterized as one of the most powerful chelating agents among natural organic substances. Their zwitter-ionic character allows them to take part in various interactions involving anions and cations. The unique chemical properties of HSs determine the possibility of their existence as buffer systems to regulate the protolytic balance in various biospheres, as well as acting as traps for free radicals [26]. The various functional groups within HS molecules exhibit the potential to function as sites for the reduction of metal ions. After the formation of AgNPs, HSs can enhance the colloidal stability of AgNPs by acting as effective capping agents [27,28].

HSs have previously been shown to exert many pharmacological effects mediated by their AOA and antiradical activity (ARA), such as hepatoprotective [29,30], neuroprotective [31], nephroprotective [32], membrane-protective [33], cardioprotective [34], and antihypoxic actions [35]. Several potential mechanisms of AOA have been described in the literature: the ability of HSs to act as proton donors; the presence of a large number of phenolic hydroxyls and quinoid structures in the structure [29,36,37]; the ability to act as free radical traps due to their high paramagnetism analogous to semiquinone-type radicals [37,38]; and the chelating of variable valence metals [39,40]. Also, it was shown that the AOA of HSs can be explained by non-phenolic fragments in the structure, including carbohydrate fragments [29].

In general, AOA is strongly connected with the manifestation of biological activity and determines the unique chemical and pharmacological potential of HSs. Today, the use of HSs as biocompatible matrices is actively developing. Various highly effective nanoparticles are introduced using HS-based matrices, resulting in highly reactive and biocompatible “nanocontainers” [41]. The main advantage of using HSs in comparison with synthetic analogs is their high detoxification activity and biocompatibility [42].

The current article describes a synergetic synthetic approach for obtaining new effective biomedical products that combine the “green” properties of natural multicharged polyelectrolytes—HSs and nanosized forms of silver.

The aim of the present work is to investigate the AOA of HS matrices (nanoparticle forming ligands) and active substances (biomaterials) based on AgNPs ultradispersed in the medium of these bioligands (HS-AgNPs).

## 2. Materials and Methods

### 2.1. Materials

The seven humic materials used in this study included a non-fractionated HS sample from peat (PHF-T3), humic acids from peat (Peat1 and Peat2), a commercial sample of fulvic acids (FA), commercial samples of sodium humates isolated from leonardite (CHP), and oxidized lignites (CHS, CHE). Descriptions of the humic materials used in this study and the corresponding silver-containing nanocomposites (HS-AgNPs) are given in Table 1.

### 2.2. Synthesis and Characterization of Silver Nanoparticles

Seven samples of HS-AgNPs were prepared in accordance with the procedure, concentrated solutions of HS (15 g/L) were used as HS matrices, AgNO_3_ was used as a precursor. In short, the silvers precursor was added dropwise to the HS solution until a concentration of 20 mmol/L was reached. The mixture was subsequently heated to 80 °C and maintained at this temperature for 4 h with continuous stirring. The obtained nanocomposites were investigated by transmission electron microscopy (TEM). TEM images were obtained using a JEOL JEM-2100F microscope (JEOL, Akishima, Japan). Image processing was performed using ImageJ software.

The morphology and particle size were analyzed for all compositions. Based on the results, the following sample composition was selected: 15 g/L HS and 20 mmol/L silver.

In all obtained samples, the form of silver and its content was calculated and confirmed. The total silver concentration was determined by the ICP-AES method using the axial ICP-AES 720-ES spectrometer (Agilent Technologies, Santa Clara, CA, USA). The UV-visible absorption spectra were measured to confirm the presence and amount of AgNPs by the maximum position and the surface plasmon resonance (SPR) peak intensity, which for AgNPs, were observed at wavelengths of 400–430 nm [43]. The UV/Vis spectra were recorded using a UV/Vis spectrometer (Cary 50 Probe, Varian, Palo Alto, CA, USA).

The data obtained for the nanosized silver content in the preparative samples were compared with the data for the total silver content to determine the degree of conversion of ionic silver into AgNPs.

### 2.3. Investigation of the HS Matrix Structure by ^13^C-NMR Spectroscopy

^13^C NMR spectra were recorded for all preparations in 0.3 M NaOD solution in D_2_O (99+% isotopic purity, Sigma Aldrich, USA). The spectra were recorded on an Avance-400 NMR spectrometer (Bruker, Berlin, Germany) with a carrier frequency for ^13^C nuclei of 100 MHz. The INVGATE pulse sequence was used to exclude the nuclear Overhauser effect. A relaxation delay of 8 s was used for the complete relaxation of quaternary carbon atoms. To calculate the structure–group composition, the spectra were divided into nine intervals corresponding to the main structural components of HS, and the intervals were integrated. The obtained integrals normalized to the whole spectrum represent the quantitative data of the structure–group composition of the studied HSs [44].

### 2.4. Total Antioxidant Capacity

The total antioxidant capacity of the HS matrices and HS-AgNPs was evaluated with the ABTS assay using the Unico 2800 spectrophotometer (Suite E Dayton, NJ, USA). The absorbance was measured at 734 nm. The studied samples interacted with a stable radical cation, ABTS^•+^ (diammonium salt of 2,2’-azino-di-(3-ethylbenzthiazoline sulfonic acid)), reducing its content in the reaction mixture [45]. The radical neutralization activity was expressed as the IC_50_—concentrations of HSs and HS-AgNPs at which the concentration of the ABTS^•+^ cation radical was reduced by 2-fold. Trolox (Acros Organics, Slovakia) was used as a positive control.

### 2.5. Cytotoxicity Study

The 3T3-L1 fibroblast cell line was obtained from the ATCC, Manassas, Virginia, USA (CL-173). The effects of the HS matrices and HS-AgNP bionanomaterials on the viability of the 3T3-L1 line was assessed using the neutral red test, as described previously [46]. Briefly, 3T3-L1 cells were cultured under standard conditions (5% CO_2_ atmosphere, DMEM/F-12 medium (GibcoTM, Billings, MT, USA) + 10% FBS (GibcoTM, Billings, MT, USA). Aqueous solutions of the tested HS and HS-AgNP samples were added in the concentration range of 7.8–1000 μg/mL. Cell plates were placed in a CO_2_ incubator for 24 h. After washing with 1 × PBS, 40 μg/mL of neutral red (Sigma Aldrich, USA) working solution was added to the wells for 2 h at 37 °C. After triple washing, the dye was extracted using neutral red destain solution (50% ethanol 96%, 49% deionized water, 1% glacial acetic acid (Sigma Aldrich, USA). The optical density was measured at 540 nm and a reference wavelength of 650 nm using a Tecan Infinite 200 pro mplex multifunctional plate reader (Tecan Group Ltd., Mannedorf, Switzerland).

### 2.6. Antioxidant/Antiradical Studies In Vitro

The production of intracellular reactive oxygen species (ROS) was assessed using the fluorescent probe 2,7-dichlorodihydrofluorescein diacetate (DCFDA). Intracellular ROS production was induced by hydrogen peroxide (H_2_O_2_). The 3T3-L1 cells were cultured under standard conditions and seeded onto black 96-well culture plates for fluorescence measurements (1 × 10^4^ cells/well). HS matrices and HS-AgNPs (12.5 μM) or Trolox (10 μM) were added to the respective wells and incubated for 24 h. Cells were then washed from samples. DCFDA working solution (10 μM) (Sigma-Aldrich, USA) was added to the corresponding wells. The plates were incubated in the thermostat for 20 min at 37 °C, washed to remove the DCFDA, and stimulated with 100 μM H_2_O_2_ prooxidant. After incubation for 60 min at 37 °C, the fluorescence in the wells was determined at λ_ex_ = 485 nm and λ_em_ = 530 nm using a Tecan Infinite200 pro mplex multifunctional plate reader (Switzerland) [47].

## 3. Results

### 3.1. ^13^C NMR Spectroscopy

The HS structure–group compositional data obtained by integrating the co-quantitative ^13^C NMR spectra over the spectral bands are summarized in Table 2.

Peat fulvic acids (FAs) and humic acids (HAs) of peat (Peat1, Peat2) are the types of natural organic matter that are the least biotransformed. These samples are characterized by having the highest contents of alkyl chain carbon (CH_n_): 27–28%. For more humified lowland peat HSs (PHF-T3), this parameter is in the same range of values as for HSs of coal: 11.5–17.5%. For all peat HSs, the carbon content of alkoxy groups (CH_n_O: polysaccharides, esters of aliphatic alcohols) is 14.5–23%, which, as expected, is higher than this parameter for HSs of coal, which has a value of 6.5–11%. According to the carbon contents of the aromatic rings for C_ar_ and C_ar_O (and the total content for both C_ar_ and C_ar_O), HSs can be divided into three groups: HSs of coal (CHP and CHS)—41–42% C_ar_ and 10.5% C_ar_O; HSs of coal and lowland peat (CHE and PHF-T3)—33–34% Car and 9.7–9.8% C_ar_O; and HSs of upland peat (Peat1, Peat2, FA)—21–24% C_ar_ and 7–8% C_ar_O. The highest content of carboxylic and ester groups (20%) is expected for FA peat, as well as for HS samples of CHS (19.4%) and CHE (17%). For the CHS sample, the COO content was 14.2%, and for peat HS and HA, it was 10.5–12%. The carbon content in carbonyl groups (C=O) for HS coal was 6–7%, and for HS peat, it was 3–5%. According to the total content of carboxyl and carbonyl carbon atoms, the studied HSs can be divided into two groups: HSs of coal and peat FAs (21–25.5% COO together with C=O) and peat HSs (14–15%). The spectral bands of acetal carbon atoms of HSs (CCA) are characterized by low signal intensities and significant overlap with the spectral band of Cag, which does not allow us to identify any stable patterns for the carbon content of CCA groups in the studied HSs (Table 2).

For better visualization of the data, they were plotted into the diagram at the coordinates “CH_n_ vs. ƩC_ar_/CH_n_O”, where ƩC_ar_ is the total carbon content of C_ar_ and C_ar_O (Figure 1). In the diagram, the analyzed HSs form three clusters: HSs of coal (CHP and CHS); HSs of coal and lowland peat (CHE and PHF-T3); HAs (Peat1, Peat2), and FAs of upland peat.

### 3.2. Characterization of Silver Nanoparticles Obtained by Transmission Electron Microscopy

From the obtained TEM images (Figure 2), the average size of the HS-AgNPs was calculated, and the results are summarized in Table 3. The content presented in Table 3 shows that HS-AgNPs synthesized in FA medium (FA-AgNPs) have the largest average particle diameters.

This fact may be due to the lower content of phenolic (C_ar_O), quinone, and semiquinone groups, which are responsible for the reduction of silver ions to HS-AgNPs. For a number of HSs isolated from coal, the diameters of the formed HS-AgNPs are in the range of 8–10 nm and their size does not differ significantly depending on the HS matrices used for synthesis. HS-AgNPs synthesized in the matrix of HAs isolated from peat (samples Peat1 and Peat2) have the smallest average diameter. Their sizes are significantly different from those of HS-AgNPs synthesized in other HS matrices. It should be noted that the morphology of the formed HS-AgNPs does not change regardless of the HS matrix used. All samples are characterized by the formation of spherical nanoparticles.

### 3.3. Investigation of the Total Antioxidant Capacity of HS Matrices and HS-AgNP Bionanomaterials Using the Stable Cation Radical ABTS^•+^

The current method is based on the ability of antioxidants to inhibit the ABTS^•+^ cation radical through proton recoil and electron transfer, resulting in a decrease in the concentration of the ABTS^•+^ cation and, consequently, in the optical density of the solution in the model system [48]. The ABTS^•+^ method is simple, accessible, and informative. It allows rapid estimation of the AOA over a wide pH range. Due to the fact that the ABTS^•+^ cation radical is a single-charge positive radical, it is soluble in both aqueous and organic solvents, and it is not affected by the ionic strength of the solution.

The obtained results are presented in Figure 3 as IC_50_ values. It was shown that the original HS matrices effectively inhibited the ABTS^•+^ cation radical across a wide range of concentrations from 3.9 to 31.2 μg/mL. CHE and PHF-T3 matrix samples were the most effective (IC_50_ values of 6.8 ± 0.1 μg/mL and 7.6 ± 0.2 μg/mL, respectively). Three HS matrices, FA (8.6 ± 0.2 μg/mL), CHP (9.0 ± 0.1 μg/mL), and CHS (9.3 ± 0.3 μg/mL), showed less activity. The Peat1 and Peat2 matrices were the least effective and were not significantly different from each other in terms of IC_50_ values (10.1 ± 0.3 μg/mL and 10.5 ± 0.3 μg/mL, respectively).

The HS-AgNP samples showed higher AOA in comparison with the original HS matrices. CHS-AgNPs and FA-AgNPs were the most active samples against the ABTS^•+^ cation radical (IC_50_ values of 4.1 ± 0.1 μg/mL and 4.3 ± 0.1 μg/mL, respectively). Three samples, CHP-AgNPs (4.7 ± 0.2 μg/mL), PHF-T3-AgNPs (5.0 ± 0.1 μg/mL), and CHE-AgNPs (5.9 ± 0.1 μg/mL), showed slightly lower efficacies. Peat2-AgNP and Peat1-AgNP samples showed the lowest ability to inhibit the ABTS^•+^ cation radical (IC_50_ values of 6.8 ± 0.1 µg/mL and 7.1 ± 0.1 µg/mL, respectively).

### 3.4. Investigation of the Cytotoxic Properties of HS Matrices and HS-AgNP Bionanomaterials In Vitro

A cytotoxicity test with the vital neutral red dye was used to evaluate the viability of 3T3-L1 after incubation with the tested samples across a wide range of concentrations (from 7.8 μg/mL to 1000 μg/mL). The cytotoxic effect of the original HS matrices was negligible, which did not allow us to calculate the IC_50_.

The ultradispersal of AgNPs in the original HS matrix resulted in a significant increase in the toxicity of composite bionanomaterials compared to the original unconjugated HS matrix. Thus, HS-AgNPs can be arranged by decreasing cytotoxic properties in the following order: CHP-AgNPs, Peat2-AgNPs, FA-AgNPs, CHS-AgNPs, CHE-AgNPs, Peat1-AgNPs, and then PHF-T3-AgNPs (Figure 4).

It is known that AgNPs often exhibit pronounced cytotoxic properties [49]. When compared with the literature data, it can be concluded that conjugation of AgNPs with the original HS matrices avoids nanoscale silver compositions with excessive cytotoxicity [49].

An analysis of the obtained data on the cytotoxicity of the original HS matrices and their nanostructured compositions containing ultradispersed silver allowed us to formulate general recommendations for the design of biologically active nanomaterials with an improved safety profile. In the future, these recommendations can be used in the development of laboratory regulations for the study of AgNP pharmacological activity.

### 3.5. Investigation of the Antioxidant/Antiradical Activity of HS Matrices and HS-AgNP Bionanomaterials In Vitro

The in vitro study of AOA/ARA was carried out on 3T3-L1 cells. The main goal was to establish the ability of HS matrices and HS-AgNPs to reduce the intracellular level of reactive oxygen species (ROS) in the presence of the prooxidant H_2_O_2_.

In vitro AOA estimation is based on the ability of DCFDA to easily penetrate into the cell cytoplasm and de-esterify under the action of intracellular esterases. 2,7-Dichlorodihydrofluorescein cannot be transported out of cells and is a weakly fluorescent agent, but in reactions with oxidants, it transforms into a strongly fluorescent product—dichlorofluorescein. When intracellular oxidant levels grow, the rate of oxidation of 2,7-dichlorodihydrofluorescein increases. Thus, the activation of intracellular oxidation is accompanied by an increase in fluorescence.

H_2_O_2_ is a reactive form of oxygen and is a strong chemical oxidizer. In the human body, H_2_O_2_ is rapidly converted to a non-selectively reactive hydroxyl radical (HO^•^) by interaction with a number of transition metal ions, of which iron is the most important in vivo [50].

As a result of the experiments, it was found that all original HS matrix samples as well as biomaterials with silver nanoparticles HS-AgNPs do not exhibit prooxidant effects in vitro. At the same time, for the basal AOA (without the addition of the prooxidant) and the AOA after stimulation by H_2_O_2_, the level of intracellular ROS production significantly differed between the studied samples (Figure 5).

In the row of the initial HS matrices used in the experiment showing the evaluation of the effect on the basal level of ROS production, the most active sample is CHE, which reduced the fluorescence intensity of DCFDA by 18.9 ± 8.1%. Then, four samples of the original HS matrices showed fairly similar levels of activity: CHP (6.7 ± 4.7%), PHF-T3 (4.9 ± 7.1%), Peat1 (4.9 ± 4.6%), and FA (4.8 ± 4.7%). The CHS and Peat2 samples were characterized by low intrinsic AOA (Figure 5). When H_2_O_2_ was used as a prooxidant, the FA (27.8 ± 3.1%), Peat1 (22.6 ± 5.7%), PHF-T3 (20.5 ± 5.3%), and CHS (17.2 ± 5.1%) samples possessed the highest AOA (Figure 5). The remaining samples had less pronounced activity against H_2_O_2_-stimulated oxidative stress.

The ranking of the HS-AgNP samples in order of decreased antioxidant properties with respect to the basal level of ROS production is as follows: CHP-AgNPs (49.8 ± 2.2%), CHE-AgNPs (38.6 ± 8.9%), PHF-T3-AgNPs (22.5 ± 6.1%), CHS-AgNPs (20.8 ± 9.6%), FA-AgNPs (20.6 ± 4.4%), Peat2 (14.3 ± 6.1%), and Peat1 (1.1 ± 8.1%) (Figure 5).

Finally, the results on the effect of HS-AgNPs against H_2_O_2_-stimulated ROS production in cells were of great interest. Here, the following samples of HS-AgNPs showed the highest antiradical activity: FA-AgNPs (47.8 ± 4.1%), CHS-AgNPs (48.5 ± 4.7%), CHP-AgNPs (32.4 ± 2.4%). PHF-T3-AgNPs (28.5 ± 3.6%), CHE-AgNPs (27.9 ± 2.9%), and then Peat1-AgNPs (26.1 ± 4.6%), which were somewhat less active in inhibiting H_2_O_2_-stimulated ROS production. And, the least pronounced activity was shown for the Peat2-AgNP sample (6.4 ± 6.3%) (Figure 5).

In general, for HS-AgNPs, a higher AOA was observed, compared to the original HS matrices.

## 4. Discussion

The undeniable advantages of “green synthesis” in obtaining nanoparticles have been noted previously. In addition to the obvious environmental advantages, “green synthesis” can also provide more pronounced biological properties (antioxidant, antitumor activity, etc.), which is due to the intrinsic properties of natural matrices [51,52]. In other words, the “green synthesis” of AgNPs can be particularly useful for creating biomedical nanomaterials with improved properties and high potential for use in medical applications.

In this paper, we present and investigate a new class of AgNPs obtained by the “green synthesis” method based on natural HSs—high-molecular-weight organic compounds that are formed, transformed, and decomposed at intermediate stages of the organic matter mineralization process [53]. The use of HSs as a matrix allows the introduction of highly efficient nanoparticles, which allows highly reactive and biocompatible “nanocontainers” to be obtained. One of the main advantages of using HSs is their high detoxification activity and biocompatibility, which makes them preferable to synthetic analogs [42].

The obtained results show that the size of the AgNP depends on the structure–group composition of the HS used for the NP synthesis. Particularly, FAs yielded the largest AgNPs, with an average diameter of 13.5 ± 6.8 nm. This outcome is attributed to FAs’ characteristics, including a reduced content of Car fragments and a diminished presence of polysaccharide fragments responsible for Ag+ reduction in comparison to other HSs used in this study. Conversely, Peat1 and Peat2 samples displayed the smallest AgNPs, characterized by average diameters of 4.6 ± 1.7 nm and 4.0 ± 2.1 nm, respectively. These samples are characterized by notably elevated levels of polysaccharide fragments that are responsible for Ag+ reduction, underscoring their role in the synthesis of smaller AgNPs.

It is known that antioxidant properties make a critical contribution to the major biological activities of HSs. Therefore, it is of particular interest to study the ability of AgNP derivatives to influence the production, biological action, and elimination of ROS in comparison with the original HSs.

The literature provides mixed evidence, as some authors have reported a higher antioxidant capacity for AgNPs compared to the original matrices [54,55]. On the other hand, many authors agree that the production of AgNPs, on the contrary, leads to pronounced prooxidant properties, and such nanoparticles can induce oxidative stress in vitro and in vivo [56,57]. The interest in this area is confirmed by the fact that a significant number of papers have been published in recent years on the study of the prooxidant/antioxidant properties of AgNPs [58].

In general, an overwhelming number of researchers agree that the ability of AgNPs to bind to free radicals and interrupt oxidation chain processes is primarily due to the antioxidant and, in particular, antiradical activity of the initial matrices used for synthesis [58]. HSs have their own pronounced AOA due to the presence of certain structural determinants in their molecular structures [29,37]. They contain a large number of fragments that are capable of oxidation. Phenolic groups are among the most important ionizable centers in the structure of HSs, determining the AOA of these compounds [59]. The antioxidant/antiradical properties of HSs are also attributed to the presence of semiquinone-type radicals in their structures [60], the stability of which is maintained by intramolecular polyconjugated aromatic systems [37] and may also be due to non-phenolic fragments in the structure, including carbohydrate fragments [29].

Many methods used to determine the antioxidant capacity of nanoparticles have been described in the literature [58]. One of the standard and generally accepted methods is the ABTS free radical scavenging method [58,61]. However, this method does not reflect the AOA of substances at the cellular level, because it does not take physiological conditions such as the pH, temperature, and bioavailability into account. Therefore, in our work, we supplemented the standard ABTS test with in vitro experiments using cell culture and the fluorescent DCFDA probe.

In our experiments, the studied AgNPs obtained using bioligands such as HSs showed a pronounced ability to bind to ROS in the test with ABTS. Moreover, in vitro tests using DHFDA showed more pronounced AOA for HS-AgNP samples compared to the original HS matrices. Silver is a catalyst of reductive reactions underlying the realization of antioxidant effects, and giving silver a nanoscale form increases its reactivity due to the dispersion properties of particles and the high ratio of their surface area to particle volume [62,63]. It was previously demonstrated that AgNPs can be used as catalytic agents in reduction reactions of methylene blue [63], the azo dyes Direct Orange 26 (DO26) and Direct Blue 15 (DB15) [64], as well as 4-nitrophenol [65]. The process of ROS inactivation using catalysts is based on electron transfer from the donor (HS matrix) to the acceptor (active oxygen radical). Silver nanocatalysts facilitate electron transfer between the nucleophilic HS molecule and the electrophilic free radical [66].

The antioxidant activity of HSs was lower than that obtained for HS-AgNPs in all ranges of HSs used for AgNP synthesis. Particularly noteworthy was the elevated AOA observed in HS-AgNPs derived from more aromatic and less oxidized coal HSs (CHP and CHS samples), as well as PHF-T3. These HS variants showcased the highest phenolic components within their structural compositions. Furthermore, FA, characterized by an increased content of -COOH groups in its molecular composition, displayed enhanced AOA. This augmentation in antioxidant potential can be attributed to the promotion of AOA in phenolic acids (especially by -CH2COOH and -CH=CHCOOH groups). Moreover, FA-AgNPs have a higher average diameter, which could also affect the AOA.

Thus, the high AOA of the tested novel HS-AgNPs bionanomaterials is most likely due to (1) the intrinsic pronounced ability of HS to inactivate ROS, (2) the large surface area of HS-AgNPs and the large surface-to-volume ratio, which make electron transfer possible and allow the kinetic barrier to be overcome for the reduction reaction. Since the reaction occurs on the surface of the nanoparticle, the increase in the surface area increases the efficiency of ROS elimination [67].

The state of imbalance between ROS formation in the body and antioxidant defense mechanisms is named “oxidative stress” and is associated with an overwhelming number of pathologies in modern humans [68]. Oxidative stress has been found to be associated with endothelial dysfunction, inflammation, hypertrophy, apoptosis, cell migration, fibrosis, angiogenesis, oncology, and neurodegenerative diseases [69,70,71].

The role of ROS and oxidative stress in the process of changing stages in wound healing deserves special attention [72]. Excessive and uncontrolled oxidative stress contributes to the dysregulation of inflammatory processes due to the oxidative modification of cell biomolecules and the induction of apoptosis, which leads to the formation of chronic non-healing wounds [73]. AgNPs are often used for the treatment of purulent and non-healing wounds infected with antibiotic-resistant microorganisms [74,75].

Considering that pronounced immunomodulatory, antiviral, and antimicrobial properties have been shown for HSs [76,77] and antimicrobial and wound healing properties of nanosized silver are well known in practical medicine, the samples of new HS-AgNP bionanomaterials based on natural HSs presented in this work could be used as wound healing agents in the future because of their non-toxicity, affordability, environmental friendliness, and high efficacy against bacteria. Therefore, the antioxidant properties of HS-AgNPs synthesized using biogenic humic matrices that we have discovered are of particular interest for studies of biological activity aimed at inhibiting the growth of bacteria and viruses and healing purulent wounds.

## 5. Conclusions

In the current article, we described the synthesis of novel silver nanoparticles, stabilized with humic macroligands (HS-AgNPs). The main aim was to evaluate the AOA of HS matrices (macroligands) and HS-AgNPs. The presented approach to obtain HS-AgNPs allows us to significantly improve the biological activity of new biomaterials by involving the “green” properties of HS. The synthesized HS-AgNPs were studied by 13C NMR and TEM. AOA plays a central and critical role in realizing the numerous beneficial biological properties of HSs. In our study, the stabilization of AgNPs with the HS matrices reduced the cytotoxicity of AgNPs in vitro. Also, HS-AgNPs demonstrated a pronounced ability to bind to ROS in an ABTS-based test of the total antioxidant capacity. It is very interesting that higher AOA was observed for HS-AgNPs versus the initial HS matrices. Another cluster of in vitro experiments involved an AOA evaluation through the DCFDA test. Similar to the ABTS test, it was found that HS materials and HS-AgNPs did not demonstrate any prooxidant effects. More pronounced AOA was shown for HS-AgNP samples. Finally, we hypothesize the involvement of two closely related mechanisms that may explain the observed phenomenon of the higher AOA activity of HS-AgNPs versus the initial matrices in both ABTS and in vitro tests. Firstly, HSs possess pronounced AOA due to the presence of fragments that are capable of oxidation. Phenolic group semiquinone-type radicals are involved in the AOA of these compounds. Secondly, the large surface area and surface-to-volume ratio of HS-AgNPs facilitate electron transfer and mitigate kinetic barriers for the reduction reaction. The examined antioxidant properties of the HS-AgNPs are of particular interest for biomedical applications in different areas.

## Figures and Tables

**Figure 1 polymers-15-03386-f001:**
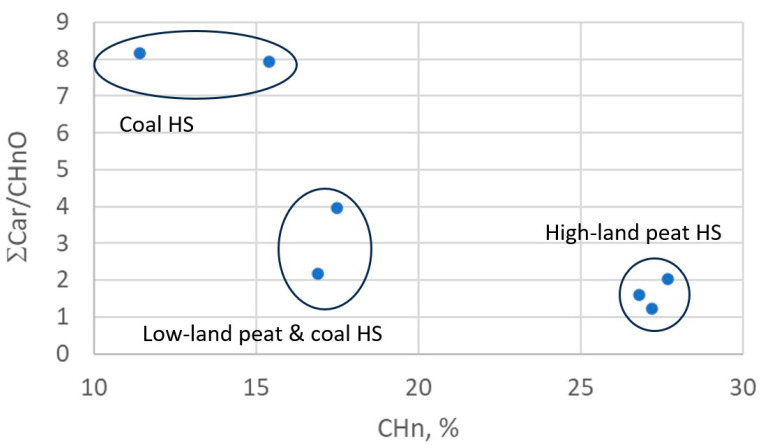
HS structure–group composition data in “CH_n_ vs. ƩC_ar_/CH_n_O” coordinates.

**Figure 2 polymers-15-03386-f002:**
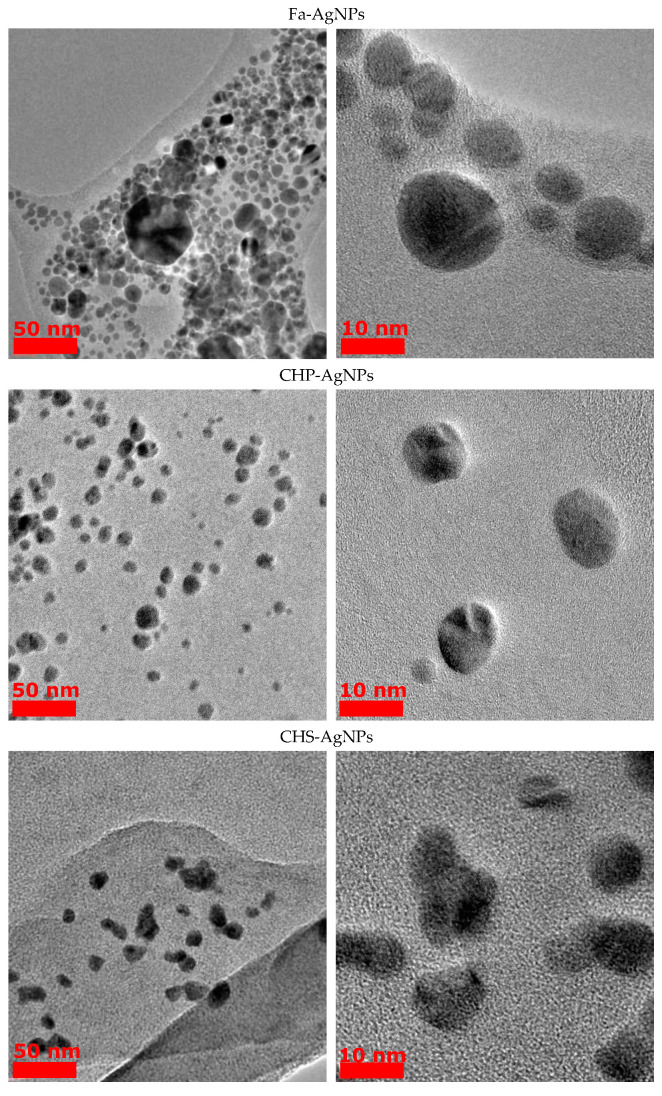
TEM images of AgNPs obtained from HS medium.

**Figure 3 polymers-15-03386-f003:**
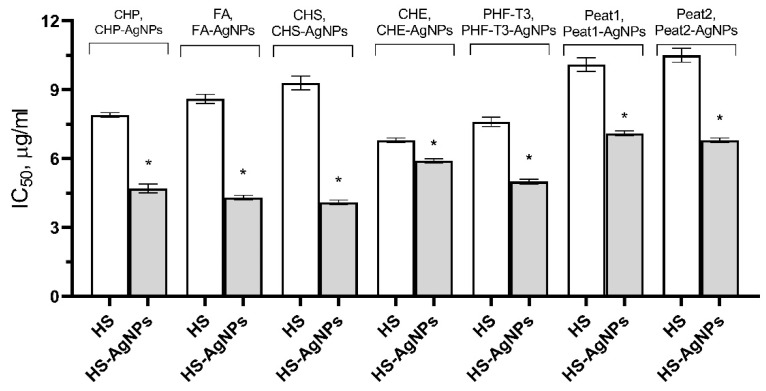
Effect of HS matrices and HS-AgNP bionanomaterials on the inhibition of the ABTS^•+^ cation radical in the model system. * Differences from the “HS” group are statistically significant, *p* ˂ 0.05.

**Figure 4 polymers-15-03386-f004:**
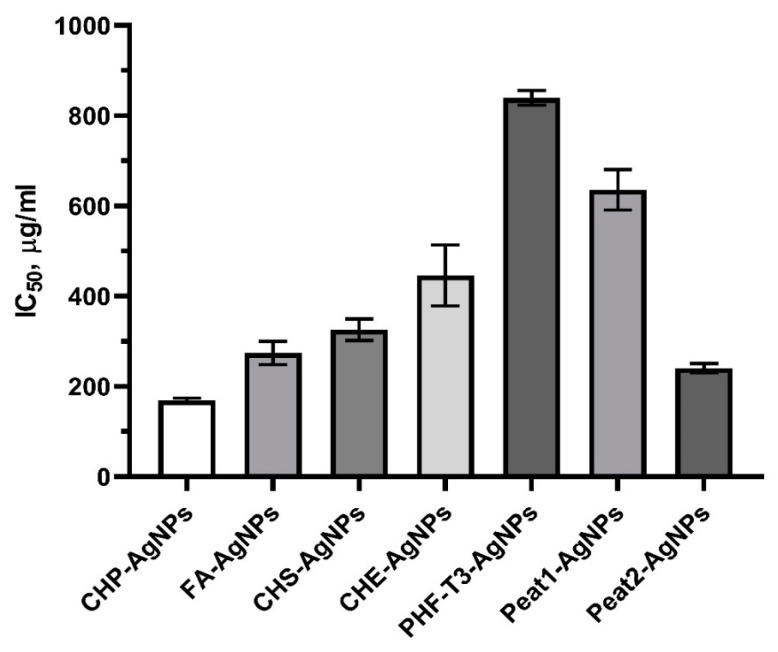
Effect of HS matrices and HS-AgNPs bionanomaterials on the viability of 3T3-L1 cells after incubation for 24 h (IC_50_, μg/mL).

**Figure 5 polymers-15-03386-f005:**
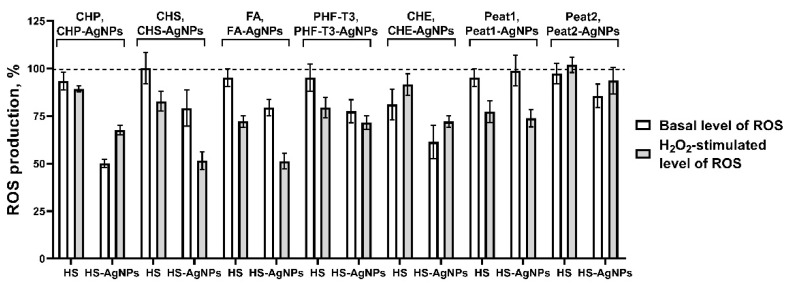
Effects of HS matrices and HS-AgNP bionanomaterials on basal (without added prooxidant) and hydrogen peroxide (H_2_O_2_)-stimulated intracellular ROS production in a cell culture of the 3T3-L1 cell line.

**Table 1 polymers-15-03386-t001:** Humic substance (HS) matrix samples and bionanomaterials based on silver nanoparticles ultradispersed in the original HS matrices (HS-AgNPs).

HS Sample Codes (Matrix)	Descriptions of HS Samples	HS-AgNP Sample Codes (with Silver Nanoparticles)
CHP	Humic acids of coal “Powhumus” (Humintech Ltd., Grevenbroich, Germany)	CHP-AgNPs
FA	Fulvic acids of fulvagra peat (Humintech Ltd., Grevenbroich, Germany)	FA-AgNPs
CHS	Humic substances of coal (Sakhalin humates LLC, Sakhalin, Russia)	CHS-AgNPs
CHE	Humic substances of coal (Genesis LLC, Novosibirsk, Russia)	CHE-AgNPs
PHF-T3	Unfractionated preparation from oligotrophic peat (Moscow, Russia)	PHF-T3-AgNPs
Peat1	Humic acids of oligotrophic angustifolium peat (Tomsk, Russia)	Peat1-AgNPs
Peat2	Humic acids of oligotrophic sphagnum-hollow peat (Tomsk, Russia)	Peat2-AgNPs

**Table 2 polymers-15-03386-t002:** Carbon distribution in the molecular structures of HS samples (matrices used in the synthesis of silver-containing bionanomaterials) according to the ^13^C NMR data.

Sample	Carbon Distribution (from ^13^C NMR), %						
	CH_n_	CH_n_O	OCO	C_ar_	C_ar_O	COO	C=O
CHP	15.4	6.5	5.4	41.1	10.5	14.2	7.0
FA	27.7	14.4	3.4	21.6	7.7	20.1	5.1
CHS	11.4	6.4	4.5	41.8	10.5	19.4	6.0
CHE	17.5	11.0	5.0	33.8	9.7	17.0	6.1
PHF-T3	16.9	19.6	6.7	32.9	9.8	10.6	3.4
Peat1	27.2	23.2	6.3	21.2	7.1	12.0	2.9
Peat2	26.8	19.8	7.2	24.1	7.7	11.6	2.8

**Table 3 polymers-15-03386-t003:** Mean diameter of HS-AgNPs synthesized in different humic matrices calculated from TEM image processing.

Sample	d ± SD, nm	n
CHP-AgNPs	10.1 ± 4.4	259
FA-AgNPs	13.5 ± 6.8	190
CHS-AgNPs	8.7 ± 6.1	284
CHE-AgNPs	7.7 ± 2.9	60
PHF-T3-AgNPs	9.5 ± 6.3	147
Peat1-AgNPs	4.6 ± 1.7	203
Peat2-AgNPs	4.0 ± 2.1	222

## Data Availability

Not applicable.

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
