# Peer review of "Enhanced Antioxidant Activity and Reduced Cytotoxicity of Silver Nanoparticles Stabilized by Different Humic Materials"

_polymers, 2023, doi:10.3390/polym15163386_

Round 1

Reviewer 1 Report

The manuscript titled 'Enhanced antioxidant activity and reduced...' has as main objective the synthesis and study of some silver nanoparticles protected by some humic materials. The subject is somehow very well known in literature but this work brings out some now insight.  After a good introduction and experimental section, the results one needs more attention, maybe to be mixed with discussion one. 72 references ends the manuscript. Based on these comments, I suggest to have a good improvement of the whole work before accepting for publication. More chemical and structural details about the humic substances will be a positive asset for the manuscript. Some other small issues:

-line 18- 'The aim s to investigate';

-text should be justified, not alignet to left;

-all tables should fit the authors instructions;

-time and temperature for the synthesis of silver nanoparticles should be provided;

-a conclusions section should be provided.

Author Response

Response to Reviewer 1 Comments

We would like to thank the distinguished reviewer for his careful study of our manuscript. We took note of all the comments, corrected the errors and edited the manuscript according to your comments.

Point 1: After a good introduction and experimental section, the results one needs more attention, maybe to be mixed with discussion one.

Response 1: We took note of this comment and expanded the discussion of the results within the framework of the research objectives. All information is presented in the part "Results".

Point 2: 72 references ends the manuscript.

Response 2: The reference to the 73rd source of literature was in the original version of the manuscript on line 415, the last paragraph. [Considering that for HS pronounced immunomodulatory, antiviral and antimicrobial properties were shown [72, 73] and antimicrobial and wound healing properties of nanosized silver are well known in practical medicine, the samples of new bionanomaterials HS-AgNPs based on natural HS presented in this work can be used in the future as wound healing agents because of their non-toxicity, affordability, environmental friendliness and high efficacy against bacteria.]

Point 3: More chemical and structural details about the humic substances will be a positive asset for the manuscript.

Response 3: We took note of this comment and the chemistry block has been expanded, 4 new references have been added in the part «Introduction».

Point 4: Some other small issues:

-line 18- 'The aim s to investigate';

-text should be justified, not alignet to left;

-all tables should fit the authors instructions;

Response 4: We have removed all these comments. We have corrected the scientific English, typos and formatting. In particular:

1) Corrected λvozb to λex.

2) Corrected 1x104 cells to 10^4.

3) Section "Abstract". Corrected – «is» instead of letter «s» (''The aim is to investigate''). After the words «As a result,» a comma is placed.

4) Section “2.1. Materials". In Table 1, the letter «s» has been added in the third column. Corrected numbering in section 2.

5) Section “Materials and methods. 2.6 Antioxidant/antiradical studies in vitro”. Removed the letter «r» at 24 hr. Corrected subscript to λex= 485 nm and λem= 530 nm.

6) Section “3.1. 13C NMR spectroscopy». In Table 2, dots are used instead of commas in numbers. In Table 2, the subscripts CHn, CHnO, Car, CarO, Car and 9.7-9.8% CarO, Car and 7-8% CarO have been corrected. And also for the caption to Figure 1. HS structure-group composition data in "CHn vs Æ©Car/CHnO" coordinates. Table 3 adds a letter «s» in the first column. Numbering changed from Figure 2 to Figure 3.

7) Section “3.3. Investigation of the total antioxidant capacity of HS matrices and HS-AgNPs bionano-materials using the stable cation radical ABTS•+”. Numbering changed from Figure 3 to Figure 4

8) Section “3.4. Investigation of cytotoxic properties of HS matrices and HS-AgNPs bionanomaterials in vitro”. DHFDA has been replaced by DCFDA. Numbering changed from Figure 4 to Figure 5.

9) Section "References". International journal of nanomedicine has been replaced by Int J Nanomedicine. Journal of food and drug analysis replaced by J Food Drug Anal. Journal of Photochemistry and Photobiology B: Biology replaced by J Photochem Photobiol B. Journal of Clinical Medicine replaced by J Clin Med.

10) Changed the font size of the affiliation co-authors.

11) Line spacing changed from 1.0 to 0.95 throughout the article.

12) First line indent changed from 0.8 to 0.75 throughout the article.

13) Changed Abstract font to 9.

14) Changed the intervals between sections before 12 pt, after 3 pt throughout the article.

15) Changed alignment of the entire text.

16) The font size of the captions for figures and tables has been changed from 10 to 9.

17) Added subsections in section 3 (3.1-3.5)

Point 5: -time and temperature for the synthesis of silver nanoparticles should be provided;

Response 5: We have added this information to the manuscript: temperature for the synthesis of silver nanoparticles 80°C and maintained at this temperature for 4 h (time).

Point 6: -a conclusions section should be provided.

Response 6: We have added section 5 «Conclusions».

Reviewer 2 Report

The authors have presented an interesting paper on the biological activity of new biomaterials combining humic substances (HS) and silver nanoparticles.

In my opinion this work is well written and needs minor revisions.

Check the English as there are a lot of mistakes: ''The aim s to investigate''

In the methods, what does λvozb mean?

1×104 cells/well should be written as x 10^4

Part 3 results, in scientific English, you usually use . and no , for writing decimals. 13.5 mg not 13,5 mg.

TEM images need visible scale bars.

Generally the paper is well written, but it needs a careful revision of the wiriting.

Author Response

Response to Reviewer 2 Comments

We would like to thank the distinguished reviewer for his careful study of our manuscript. We took note of all the comments, corrected the errors and edited the manuscript according to your comments.

Point 1: Check the English as there are a lot of mistakes: ''The aim s to investigate'' In the methods, what does λvozb mean? 1×104 cells/well should be written as x 10^4. Part 3 results, in scientific English, you usually use. and no , for writing decimals. 13.5 mg not 13,5 mg.

Response 1:

We have removed all these comments. We have corrected the scientific English, typos and formatting. In particular:

1) Corrected λvozb to λex.

2) Corrected 1x104 cells to 10^4.

3) Section "Abstract". Corrected – «is» instead of letter «s» (''The aim is to investigate''). After the words «As a result,» a comma is placed.

4) Section “2.1. Materials". In Table 1, the letter «s» has been added in the third column. Corrected numbering in section 2.

5) Section “Materials and methods. 2.6 Antioxidant/antiradical studies in vitro”. Removed the letter «r» at 24 hr. Corrected subscript to λex= 485 nm and λem= 530 nm.

6) Section “3.1. 13C NMR spectroscopy». In Table 2, dots are used instead of commas in numbers. In Table 2, the subscripts CHn, CHnO, Car, CarO, Car and 9.7-9.8% CarO, Car and 7-8% CarO have been corrected. And also for the caption to Figure 1. HS structure-group composition data in "CHn vs Æ©Car/CHnO" coordinates. Table 3 adds a letter «s» in the first column. Numbering changed from Figure 2 to Figure 3.

7) Section “3.3. Investigation of the total antioxidant capacity of HS matrices and HS-AgNPs bionano-materials using the stable cation radical ABTS•+”. Numbering changed from Figure 3 to Figure 4

8) Section “3.4. Investigation of cytotoxic properties of HS matrices and HS-AgNPs bionanomaterials in vitro”. DHFDA has been replaced by DCFDA. Numbering changed from Figure 4 to Figure 5.

9) Section "References". International journal of nanomedicine has been replaced by Int J Nanomedicine. Journal of food and drug analysis replaced by J Food Drug Anal. Journal of Photochemistry and Photobiology B: Biology replaced by J Photochem Photobiol B. Journal of Clinical Medicine replaced by J Clin Med.

10) Changed the font size of the affiliation co-authors.

11) Line spacing changed from 1.0 to 0.95 throughout the article.

12) First line indent changed from 0.8 to 0.75 throughout the article.

13) Changed Abstract font to 9.

14) Changed the intervals between sections before 12 pt, after 3 pt throughout the article.

15) Changed alignment of the entire text.

16) The font size of the captions for figures and tables has been changed from 10 to 9.

17) Added subsections in section 3 (3.1-3.5)

18) Added section 5. Conclusions.

Point 2: TEM images need visible scale bars.

Response 2: We added rulers to drawings (TEM images visible scale bars). And we sent the raw data of the TEM images in Figure 2 with the original scale bar (zip file) for the editor.
